# Urticaria: A Narrative Overview of Differential Diagnosis

**DOI:** 10.3390/biomedicines11041096

**Published:** 2023-04-04

**Authors:** Natale Schettini, Monica Corazza, Cecilia Schenetti, Lucrezia Pacetti, Alessandro Borghi

**Affiliations:** Section of Dermatology and Infectious Diseases, Department of Medical Sciences, University of Ferrara, 44121 Ferrara, Italy

**Keywords:** urticaria, differential diagnosis, auto-inflammatory syndromes, autoimmune disorders, drug-related eruptions, hyperproliferative diseases

## Abstract

Urticaria is an inflammatory skin disorder that may occur in isolation or associated with angioedema and/or anaphylaxis. Clinically, it is characterized by the presence of smooth, erythematous or blanching, itchy swelling, called wheals or hives, which greatly vary in size and shape and last less than 24 h before fading to leave normal skin. Urticaria is the consequence of mast-cell degranulation that can be caused by immunological or non-immunological mechanisms. From a clinical point of view, many skin conditions can mimic urticaria and their recognition is mandatory for a correct management and therapeutic approach. We have reviewed all of the main relevant studies which addressed differential diagnosis of urticarial, published until December 2022. The National Library of Medicine PubMed database was used for the electronic research. The present review offers a clinical narrative overview, based on the available literature, of the principal skin disorders that can be misdiagnosed as urticaria (mainly autoinflammatory or autoimmune disorders, drug-induced reactions, and hyperproliferative diseases). The aim of this review is to provide clinicians a useful tool for correctly suspecting and identifying all of these conditions.

## 1. Introduction

### 1.1. Definition, Classification and Pathogenesis of Classic Urticaria

Urticaria is an inflammatory disorder presenting with transient wheals, angioedema or both, without systemic symptoms. It is the consequence of the degranulation of mast-cells that can be spontaneous or triggered by various agents. Acute urticaria (duration < 6 weeks) often associates with infections, or food or drugs intake. Chronic urticaria (CU) (duration > 6 weeks) may be spontaneous (chronic spontaneous urticaria, CSU) when no specific trigger can be found, associated with systemic disorders or inducible (chronic inducible urticaria, CIndU) by well-defined stimuli [1,2].

The pathogenesis of CSU involves several factors including a genetic predisposition and undefined triggers such as drugs, infections, systemic diseases and allergy. These agents can induce the stimulation of autoimmunity pathways, both IgE or IgG/IgM-mediated, or the activation of the coagulation cascade that are responsible of mast-cell degranulation [3].

Depending on the type of the stimulus, CIndU can be divided into physical urticaria that may be due to stroking, rubbing or scratching in case of urticarial dermographism, cold, heat, vibration, delayed pression or sun exposure, and into non-physical urticaria including contact, acquagenic and cholinergic urticaria when provoking substances, water or rapid increase of body temperature represent respectively the specific trigger. Characteristic of CIndU is the ability to reproduce urticarial reactions with challenge tests involving the application of the suspected eliciting stimulus [4].

### 1.2. Epidemiology

CU is a common disease affecting a considerable part of the population with important regional differences. The estimated prevalence worldwide differs across studies, ranging from 0.08% to 3.08% [5,6,7].

In a recent cross-sectional analysis, CU showed a standardized prevalence of 0.23% among adults in the United States with women presenting twice the percentage of men [8]. On the contrary, in the largest European study a prevalence of 0.51% was found (although these data were not standardized) [9].

Regarding children, few epidemiologic data are available but recent studies showed a prevalence ranging from 0.7% to 1.38% [10,11].

### 1.3. Clinical and Histologic Aspects

Wheals are the main clinical manifestation of urticaria and consist in skin lesions of variable sizes and with a round or polycyclic morphology, characterized by edema and variable erythema. They are typically itchy, present an asymmetrical distribution and resolve within 24 h without any sequels. Angioedema is a painful edema of the deep dermis or subcutaneous tissues that resolves in less than 72 h.

Histology of CU shows swelling of the superficial dermis associated with dilation of vessels without leukocytoclasia or wall damage. A perivascular eosinophilic and neutrophilic infiltrate can be observed whereas macrophages or lymphocytes are less represented [12].

### 1.4. Therapy

The key points of urticaria therapy are the avoidance of known triggers and aggravating factors, patient’s reassurance and education. However, in the majority of patients, triggers are not identifiable and systemic medications are needed. These can be stratified into first-, second-, and third-line therapies [13]. The first step consists of second-generation H1 antihistamines taken on daily basis. They specifically antagonize the peripheral H1 receptors and are less sedating than first generation H1 antihistamines. H1 antihistamines reduce pruritus, flatten wheals, shorten wheal duration and reduce wheal numbers. If the first step fails, the second step involves increasing the dosage of the H1 antihistamine already in use, up to four times the recommended dose [14]. After two to four weeks, if no clinical improvement occurs, omalizumab should be considered. Omalizumab is a monoclonal antibody (recombinant humanized IgG1κ) that can prevent IgE interaction with its (FcεRI) on basophils and mast cells by binds to the constant region of free IgE [15]. The functioning of omalizumab is not fully understood; the proposed mechanisms are the following: reduction in circulating free IgE levels, reduction in expression of IgE FcεRI receptors on mast cells and basophils, decrease in mast cell sensitivity, reduction in IgE–basophil crosslinking, reduction in IgG autoantibody activity. At the moment omalizumab is the only monoclonal antibody approved for chronic spontaneous urticaria in patients aged 12 or more at the dosage of 300 mg every four weeks for at least six doses.

Several guidelines consider as ciclosporin as a possible third-line therapy for chronic spontaneous urticaria [16]. Cyclosporine (2–4 mg/kg/day) appears to benefit approximately two-thirds of patients with chronic autoimmune urticaria refractory to antihistamines, however in most patients the remission is only temporary, and relapse occurs after few months [17]. Even if cyclosporine fails other immunosuppressive therapies might be considered such as azathioprine, methotrexate and micophenolate mofetil. Regarding the use of systemic corticosteroids, short-term therapy (i.e., maximum of 10 days) can be used in acute urticaria or exacerbations of chronic spontaneous urticaria [18].

### 1.5. Differential Diagnosis

There are several diseases, mainly autoinflammatory or autoimmune disorders, drug-induced reactions and hyperproliferative diseases, that may mimic urticaria and that must be distinguished for a correct therapeutic approach. Frequently these conditions are characterized by the development of atypical wheals that may be infiltrated or have a duration exceeding 24 h with a scaly or a hypo/hyperpigmented evolution. These wheals may present a symmetric distribution and be associated with papules, vesicles, bullae or other elementary skin lesions, but usually with no angioedema. Another important sign of these disorders is the presence of systemic symptoms such as unexplained recurrent fever, joint or bone pain and malaise [19,20].

Conventional therapies for chronic spontaneous urticaria are usually not effective especially in autoinflammatory fever syndromes and neutrophilic urticarial dermatoses. The former group responds to anti-IL-1 agents such as anakinra, while the latter is usually responsive to dapsone or colchicine [16].

The main differential diagnoses of urticaria are summarized in Table 1.

The aim of this review is to offers a clinical overview, based on the available literature, of the principal skin disorders that can be misdiagnosed as urticaria and provide clinicians a useful tool for correctly suspecting and identifying all of these conditions.

## 2. Autoinflammatory Urticarial Syndromes

Autoinflammatory urticarial syndromes (AUS) represent a heterogeneous group of disease that may be hereditary or acquired. Dysregulation of IL-1 pathway and hyperactivation of inflammasome are suspected to be the main pathogenetic mechanisms underlying these disorders [21].

Diagnostic criteria include urticarial lesions and several systemic symptoms. In AUS eruptions are typically characterized by flat and only slightly itchy wheals lasting up to 24 h and mainly involving extremities and trunk. Laboratory shows elevated inflammation markers whereas neutrophilic leukocytosis can be found on skin biopsy [19,22].

The main AUS and their characteristics are summarized in Table 2.

### 2.1. Neutrophilic Urticarial Dermatosis

Neutrophilic urticarial dermatosis (NUD) is a rare condition characterized clinically by an urticarial rash and histologically by an intense neutrophilic infiltrate. It may have a chronic or a recurrent course interspersed with symptom-free periods. The skin eruption occurs especially on the trunk and the limbs and is characterized by erythematous macules and papules, sometimes confluent in barely raised plaques, with less pronounced itch and edema than classic urticaria. This rash generally presents a longer duration with a resolution occurring within 24 to 48 h. Purpuric aspect or hyperpigmented sequels are not observed.

Systemic symptoms are frequent and typically represented by fever, fatigue, myalgia and arthralgia; chest or abdominal pain can also be reported [23]. These extra-cutaneous signs can be present even with no underlying disorders. In fact NUD is often associated with systemic conditions, such as cryopyrin-associated periodic syndrome adult-onset Still disease or systemic lupus erythematosus, but it can occur also as an entity on its own.

Histology of NUD shows a dermal neutrophilic infiltrate with interstitial and perivascular spreading and leukocytoclasia. Neutrophils are typically arranged in Indian file next to collagen bands and in sweat glands can present an epithelial tropism. NUD differs from urticaria vasculitis for the absence of fibrinoid necrosis of vessel walls and from Sweet syndrome since no significant dermal edema can be found [24].

### 2.2. Cryopyrin-Associated Periodic Syndrome

Cryopyrin-associated periodic syndrome (CAPS) represents a group of autosomal dominant autoinflammatory diseases due to gain-of-function mutations in the nucleotide-binding oligomerization domain-like receptors (NLR), family pyrin domain containing 3 (NLRP3) genes. These mutations are responsible for a hyperexpression of pro-inflammatory cytokines, such as IL1β and IL18, caused by a constant activation of the NLRP3 inflammasome [22].

The CAPS spectrum encompasses three overlapping phenotypes presenting an increasing impact on patients’ clinical conditions: familial cold auto-inflammatory syndrome (FCAS), Muckle-Wells syndrome (MWS), and chronic infantile neurologic, cutaneous and articular syndrome (CINCA), also called neonatal-onset multisystem inflammatory disease (NOMID). These disorders typically present an early onset in the first few years of life and are characterized by systemic symptoms such as recurrent fever, fatigue, joint and muscular pain and variable organ involvement.

Skin is frequently affected by a non-itchy or only slightly pruritic urticarial eruption characterized by symmetrical distribution of macules, papules or plaque with a pink or pale red color. A halo of vasoconstriction may be observed at the periphery of the lesions whereas edema or annular configuration are not reported. Cutaneous eruptions in CAPS have a longer duration than classic urticaria (24–48 h) but still resolve with no sequels.

In FCAS, skin rashes often appear 2 h after cold exposure, especially in areas not directly exposed such as the upper arms, trunk, buttocks and thighs but may resolve spontaneously within 24 h. After 4–6 h, other symptoms such as fever and arthralgia may occur. Nausea, vomiting and conjunctivitis are also reported. MWS is characterized by a longer lasting skin eruption (one to three days) and a higher frequency of progressive sensorineural hearing loss and amyloidosis than FCAS. CINCA/NOMID, the most severe phenotype, usually presents an early onset with congenital and often continuous urticarial eruption associated with neurologic, ocular and joint involvement that are responsible for the low life expectancy of this disorder [21]. Histology of CAPS typically presents the pattern of NUD.

There are also other autosomal dominant auto-inflammatory diseases that may present with urticarial rashes. These conditions are caused by mutations in various genes such as the NLR family pyrin domain containing 12 (NLRP-12), the phospholipase C gamma 2 (PLCG2) and the NLR family caspase recruitment domain containing 4 (NLRC4) [22].

### 2.3. Schnitzler’s Syndrome

Schnitzler’s syndrome is a rare condition considered a paradigm of acquired auto-inflammatory disorder with late onset (mean age 50 years). Its etiology is still unclear even if a hyperactivation of IL1 pathway is supposed, considering the efficacy of IL1-inhibitors in the treatment of the disease.

The clinical features of Schnitzler’s syndrome are represented by an urticarial eruption usually less itchy and longer lasting (12 to 36 h) than classic urticaria and characterized by lesions that in some cases resolve with a brown hyperpigmentation. Systemic symptoms include an otherwise explained recurrent fever > 38 °C that typically appears with the cutaneous eruption, fatigue or general malaise, arthralgia, myalgia or bone pain. Lymphadenopathy, liver and/or spleen enlargement are also reported. The main laboratory finding is a monoclonal IgM gammopathy that represents an obligate criterion for the diagnosis together with the chronic urticarial rash. A few cases with a monoclonal IgG gammopathy are also described. Further reported features include leukocytosis and elevated sedimentation rate [25].

A neutrophilic dermal infiltrate, identical to those described in NUD, is observed on skin biopsy.

Schnitzler’s syndrome can be complicated by systemic amyloidosis or by an evolution into a lymphoproliferative disorder that can be seen in 15% of patients.

### 2.4. Adult-Onset Still’s Disease

Adult-onset Still’s disease (AOSD) is a rare systemic condition of undefined etiology that seems to be linked to a genetic predisposition, viral and bacterial infections, and a dysregulation of immune functions involving both auto-inflammatory and auto-immune pathways [26]. The onset of the disease usually occurs between the ages of 16 and 35 with a slightly higher predominance in females.

The main clinical features of AOSD include a high-spiking fever, typically with temperature ≥ 39 °C, occurring in the afternoon or early evening and often associated with an evanescent and mildly itchy skin rash characterized by salmon-pink macules and papules on the proximal limbs and trunk. The cutaneous eruption follows the course of the fever resulting more evident during the spike and disappearing with the fever resolution. In some patients other different skin manifestations have been reported such as a longer-lasting rash with a duration that reaches days or weeks without changes, plaques and linear pigmentation or urticarial eruption with symmetrical distributed atypical wheals characterized by a persistence longer than 24–36 h [27]. Other typical features of AOSD are arthralgia or arthritis, lasting at least two weeks, and leukocytosis with a marked blood neutrophilia probably linked to a granulocyte hyperplasia in the bone marrow. Minor criteria of the disease include cardio-pulmonary, liver or spleen involvement, sore throat and lymphadenopathy. According to Yamaguchi criteria, usually applied to classify and diagnose AOSD, infections, malignancies and other rheumatic diseases must be excluded [28].

Histology of cutaneous lesions, especially salmon-pink rashes, shows a perivascular, mixed lymphocytic and neutrophilic infiltrate of the superficial dermis without epidermis involvement. Necrosis of keratinocytes in the upper half of the epidermis associated with neutrophilic inflammation in the papillary dermis can be observed in case of persistent eruption with papules and plaques.

In cases of atypical urticarial lesions, histologic findings of NUD are reported [29].

### 2.5. Systemic-Onset Juvenile Idiopathic Arthritis

Systemic-onset juvenile idiopathic arthritis (SoJIA) represents an inflammatory systemic disorder largely due to a dysregulation in innate immunity pathways, especially involving pro-inflammatory cytokines such as IL-1, IL-6, and IL-18. However recent evidences have shown an important role of adaptive immunity as well, considering the activity of IL-17 producing effector CD4+ cells in the chronic stage of the disease [30].

SoJIA occurs in children < 16 years of age with high-spiking fever of unknown origin and poly-arthritis that may be associated with skin eruption, enlargement in liver, spleen and lymph nodes or serositis [31]. Cutaneous manifestations consist in a transient pink macular eruption following the course of daily fever spike and presenting the same clinical features as in AOSD.

### 2.6. Mevalonate Kinase Deficienc

Mevalonate kinase deficiency (MKD) is a syndrome caused by a mutation in both copies of the MVK-gene, responsible for a block of the mevalonate metabolism that leads to an autoinflammatory disorder.

MKD includes two phenotypes of increasing severity, named Hyper-IgD and periodic fever syndrome (HIDS) and mevalonic aciduria (MA) [32].

HIDS usually occurs in infancy (mean age at onset of 6 months) with recurrent fever, often exceeding 40 °C, associated with abdominal pain, vomiting and/or diarrhea, lymphadenopathy, arthralgia, and non-erosive arthritis. Cutaneous manifestations are heterogeneous including evanescent maculo-papular rashes, sometimes with vasculitis aspects, oral aphthous ulcers, and occasionally erythema nodosum [33].

In MA the onset of the disease may be at birth or ante-natal with congenital malformations. Neurologic alterations and hematologic abnormalities, often responsible for early death, are reported. Laboratories typically show elevated IgD levels [34].

### 2.7. TNF-Receptor-Associated Periodic Syndrome

TNF-receptor-associated periodic syndrome (TRAPS) is an auto-inflammatory condition due to pathogenic variants of the gene encoding the TNF-receptor superfamily 1A. It is a hereditary periodic fever disorder presenting in children < 10 years of age with episodes of high temperature that last from 5 to 25 days and that occur every 4–6 weeks.

Other clinical features include abdominal and chest pain, arthralgia, myalgia, eye involvement with uveitis or conjunctivitis and cutaneous manifestations [35].

The skin can be affected by migrating erythema with centrifugal evolution or less frequently by urticarial plaques [36]. Histology shows a monocytic and lymphocytic infiltrate with perivascular distribution [37].

## 3. Immune-Mediated Conditions

Immune-mediated conditions are a heterogeneous group of disorders that result from the abnormal activity (e.g., extreme inflammatory response) of immune cells due to various triggers.

### 3.1. Urticarial Dermatitis

Urticarial dermatitis (UD) is an inflammatory disorder that combines overlapping urticarial and eczematous features that develop simultaneously or sequentially.

Etiology of DHR may be linked to drug hypersensitivity, viral infections, infestations and several other clinical scenarios [38], but there are also cases of UD that can only be classified as an independent entity. In the literature cases of paraneoplastic skin eruption described as UD are also reported [39].

It occurs especially in adult patients (mean age 60 years) presenting with both erythematous, excoriated and scaly patches resembling eczema and a wheal-and-flare itchy reaction with papules and plaques mimicking urticaria but lasting longer than 24 h [40].

The histology of UD shows the so-called dermal hypersensitivity reaction (DHR) pattern that is characterized by a lymphocytic infiltrate with a superficial to mid-dermal perivascular localization and accompanied by scattered eosinophils. Only minimal epidermal spongiosis can be observed whereas stratum corneum is typically unaffected and exocytosis is less significant than eczema. This pattern differs from classic urticaria where lymphocytic inflammation extends beyond the superficial dermis involving deeper vessels as well and is associated with neutrophilic infiltrate and interstitial edema [41,42].

### 3.2. Urticarial Vasculitis

Urticarial vasculitis (UV) is a condition clinically characterized by wheals lasting more than 24 h. From a histopathological point of view, it shows leukocytoclastic vasculitis of small vessels [43]. UV is thought to be a type III hypersensitivity reaction, mediated by the formation of immune-complexes [44].

Recognizing UV is crucial, given a possible association with systemic features, however it can be challenging as the lesions of CSU and UV may be visually indistinguishable.Classical indurated wheals that last >24 h, often persist for several days, and may leave a residual ecchymotic hyperpigmentation that is not seen in urticaria (Figure 1).

The urticaria-like lesions are usually dark-red or brown, especially in the center. The erythema usually does not subside when exerting pressure. UV lesions are usually smaller than those of urticaria, ranging from 0.5 to 5 cm in diameter. The itching sensation associated with urticaria lesions is not present in UV; patients usually complain of a painful and/or burning sensation [44].

UV lesions usually resolve with residual hyperpigmentation.

A recent study shed light on a novel dermoscopic sign that can significantly aid in the diagnosis of UV: subclinical purpuric patches/globules (PG) can be seen in UV lesions with dermoscopy (19-fold increase in the odds for UV at multivariate analysis) [45].

A skin biopsy, however, is the only tool for a definitive diagnosis of UV.

The main histological criteria include: signs of karyorrhexis (nuclear dust), presence of erythrocytes outside the vessels, presence of fibrin inside the vessels and neutrophilic infiltrate. An urticarial vasculitis score (UVS), which is a combined quantitative assessment of the three criteria leukocytoclasia, fibrin deposits and extravasated erythrocytes, showed promising results in distinguishing UV from CSU in skin histopathology [45].

### 3.3. Cutaneous Lupus Erythematosus

Lupus erythematosus (LE) is a connective tissue disorder that primarily occurs in females between the third to fourth decade of life [46].

Autoantibodies, immune complexes, and loss of immune tolerance are at the base of the pathogenesis of the disease [47].

Cutaneous lupus erythematosus (CLE) is divided in three categories: chronic cutaneous lupus erythematosus (CCLE), subacute cutaneous lupus erythematosus (SCLE) and lastly acute cutaneous lupus erythematosus (ACLE) [46,47]. In ACLE an erythematous rash localized to the nose and cheeks, symmetrically (malar rash) is typical. SCLE lesions are usually localized in photoexposed areas of the body. Morphologically, SCLE lesions can be papulosquamous or annular with central clearing; both forms can occur together in the same patient and typically heal without scarring [47]. CCLE includes many forms, such as discoid lupus erythematosus (DLE), lupus tumidus, and more. DLE usually presents with well-defined, annular erythematous patches or plaques with follicular hyperkeratosis. The lesions expand, leaving depressed central atrophy and can progress to irreversible scarring alopecia on the scalp [48].

From a hystopathological point of view, all types of specific CLE manifestations share similar/overlapping features, such as: (i) epidermal vacuolization and apoptosis; (ii) an interface dermatitis; (iii) dermal inflammation; (iv) follicular plugging; (v) higher thickness of the basal membrane; and (vi) mucin deposition in the dermis [46].

Immunofluorescence studies show different patterns depending on the form in analysis and can also be described as a spectrum/overlap between entities.

Nonspecific cutaneous manifestations are even more polymorphic and can include an enormous variety of entities: photosensitivity, urticaria, erythema, vasculitis, Raynaud’s phenomenon, bullous lesions, non-scarring alopecia, oral ulcers, and more [49,50,51].

## 4. Autoimmune Disorders

Autoimmune disorders are a broad group of conditions that occur when the immune system attacks healthy cells/tissue due to the formation of auto-antibodies or T lymphocytes directed against autoantigens.

### 4.1. Bullous Pemphigoid

Bullous pemphigoid (BP) is the most common autoimmune sub-epidermal blistering disorder and usually affects elderly patients [52].

BP is immunologically characterized by tissue-bound and circulating autoantibodies directed against either the BP antigen 180 or the BP antigen 230 [53].

The clinical presentation of bullous pemphigoid can be polymorphic, but it characteristically presents with intense generalized pruritus and tense bullae containing clear fluid and evolving in eroded and/or crusted areas [52].

In typical cases, the vesicles and bullae occur on erythematous skin, together with urticarial papules and plaques that occasionally assume an annular pattern (Figure 2).

In atypical cases, or during the prodromal phases, bullous lesions may be absent; eczematous, popular, and/or urticarial cutaneous lesions can be seen instead [53].

From a histopathological point of view we can find: (i) subepidermal rupture; (ii) superficial perivascular infiltrate made up of mainly eosinophils; (iii) eosinophilic spongiosis. Spongiosis and an eosinophilic infiltrate at the papillary derma without vesiculation can also be found in urticaria, which can make early diagnosis of PB difficult.

DIF is the gold standard for diagnosis of autoimmune blistering disorders. In PB a linear deposition of C3 and IgG at the basal membrane can be found. BP immunoreactants localize to the epidermal side of the preparation, while in other autoimmune bullous diseases such as epidermolysis bullosa acquisita they localize on the dermal side.

The research of circulating auto-antibodies against BP180 and BP230 through ELISA testing may be performed to further corroborate the diagnosis [53].

### 4.2. Pemphigoid Gestationis

Pemphigoid gestationis (PG) is a rare (incidence: one case per 2000–60,000 pregnancies) autoimmune bullous dermatosis specific of pregnancy and the post-partum period [54].

Clinically PG manifests with severe pruritus and polymorphic inflammatory skin lesions: eczematous or erythema multiforme-like lesions, urticaria-like lesions, papules that can evolve into tense blisters/bullae. The rash usually start in the periumbilical area and can spread to the whole abdomen and limbs; the mucosae are spared [55,56].

PG occurs most commonly during the last trimester of pregnancy and is usually self-limiting in the first two months after delivery. More rarely it can occur in the immediate post-partum period or during the second trimester. In recurs in every following pregnancy and the onset is earlier and with a worse presentation.

Risks for the fetus can be low birth weight and prematurity; skin lesions can be found on the child at birth but resolve da sole after a few weeks from birth.

Lesions are classically sub-epidermal. At histopathology we find: (i) papillary oedema; (ii) polymorphic inflammatory infiltrate in the dermis; (iii) subepidermal blistering. IFD is the gold standard, showing linear deposition of complement and, less commonly, IgG depositions along the dermal-epidermal junction. ELISA test for BP180 is also very sensitive and specific for PG diagnosis [57,58].

Treatment depends on the severity of PG. For localized disease, topical corticosteroids and oral antihistamins can be sufficient. Systemic corticosteroids can be used in diffuse disease (>10% body surface) or for those cases resistant to topical treatment [54].

### 4.3. Dermatitis Herpetiformis

Dermatitis herpetiformis (DH) is an autoimmune bullous disorder caused by gluten. It manifests as a recurring papulo-vesicular rash, defined as extremely pruritic from the patient. The lesions can be found predominantly at the extensor surfaces of the limbs [59].

The development of IgA autoantibodies against transglutaminases are at the base of pathogenesis for both DH and celiac diseases (CD). In DH, the deposit of autoantibodies can be found in the superficial papillary dermis [60]. DH is currently regarded as a special, distinct form of CD with combined intestinal and cutaneous manifestations [61].

The disease is characterized by intensely pruritic, polymorphic rash with small blisters and papules. DH lesions affect most frequently the extensor surfaces of the elbows, knees and buttocks. More rarely, the scalp, face, upper back and neck, can be affected. Clinically, the symmetric involvement of extensor surfaces may guide the diagnosis.

The intense pruritus associated with the disease often lead to intense scratching that can break the small blisters, turning them into erosions and crusts; in the long run, this can lead to scarring. On the other hand, the absence of pruritus can strongly suggest a different diagnosis [59].

The diagnosis of DH is based on: (i) clinical picture; (ii) serology; (iii) histopathology; and (iv) direct immunofluorescence. DIF is the gold standard for diagnosis, showing deposits of IgA in the dermal papillae and/or dermoepidermal junction, with granular pattern [60].

From a histopathological point of view, an intact vesicle shows sub-epidermal split with neutrophilic infiltrate at the tips of the dermal papillae. Perivascular polymorphic inflammatory infiltrate is also common [62].

### 4.4. Autoimmune Progesterone Dermatitis

Autoimmune progesterone dermatitis (APD) is a rare skin disease that affects women of childbearing age; to date only 200 cases have been reported in the literature. APD is probably caused by exposure to exogenous and/or endogenous progesterone causing hypersensitivity reaction leading to the development of the clinical manifestation of the disease [63]. The clinical presentation of APD can include both cutaneous and non-cutaneous features. The skin presentation is extremely heterogenous: urticaria-like manifestations, vesiculobullous eruption, erythema multiforme, eczema, maculopapular eruption and purpura/petechiae. The most often reported symptom is pruritus [64].

The lesions typically appear 3–10 days prior to menses, coinciding with the luteal phase of the menstrual cycle, and resolve after menstruation. The clinical presentation is quite similar at each recurrence. Other than the association with the menstrual cycle, cases associated with pregnancy or exposure to hormone therapy have been reported [65].

## 5. Hyperproliferative Diseases

Hyperproliferative diseases are characterized by uncontrolled proliferation of cells which can be determined by various factors, from environmental triggers to genetic mutations.

### 5.1. Mastocytosis

Mastocytosis encompasses a heterogeneous group of disorders characterized by the proliferation of clonal mast cells into different tissues. Typically, these diseases are divided into cutaneous mastocytosis (CM), when only the skin is affected, and systemic mastocytosis (SM), when there is an involvement of bone marrow and other organs such as lymph nodes, spleen and gastroenteric tract. CM usually occurs in childhood and tends to resolve spontaneously after a few years or during adolescence [66,67]. Different subvariants of CM are described on the basis of their clinical aspects.

Maculo-papular CM (MPCM), formerly called urticaria pigmentosa, is characterized by yellowish to deep brown pigmented lesions that are small and round in the monomorphic variant, which usually affects adults, while they present different size and morphology in the polymorphic one, which is more frequent in children (Figure 3).

MPCM usually affects the trunk and extremities while sparing the face, scalp and palmar and plantar areas [68]. In some cases vesicle at the center of the lesions and symptoms of mast cell activation, such as flushing and pruritus, can be observed [69]. A typical feature of MPCM is the positivity of Darier’s sign which results highly suggestive for the diagnosis. 

It represents the elicitation of an urticarial reaction with the appearance of erythema and wheals after the mechanical irritation of the skin; occasionally a spontaneous urtication of the lesions is described as well. 

Telangiectasia macularis eruptiva perstans (TMEP) is characterized by multiple erythematous and hyperpigmented macules with telangiectasias [70], but without Darier’s sign. This form usually affects adults and is considered a subvariant of MPCM [66].

Cutaneous mastocytoma typically affects children within three months of life presenting with a solitary yellowish to brown macule, nodule or plaque on the extremities with positivity of Darier’s sign. Spontaneous resolution during childhood is commonly observed. 

Diffuse cutaneous mastocytosis (DCM) usually presents at birth or during early infancy with generalized erythema and pachydermia occasionally associated with marked dermographism. Blistering with possible hemorrhagic evolution is common after irritation of the skin [71]. 

CM is to be suspected when Darier’s sign is present but the definite diagnosis requires a skin biopsy with immunohistochemical staining for tryptase and KIT [72].

Biopsy of CM shows multifocal infiltrates of mast cells that present a perivascular distribution in papillary dermis and are located in skin appendage in the case of MPCM [73]. Extension to reticular dermis and subcutis is common in cutaneous mastocytoma, whereas an entire involving of the skin layers is described in DCM.

### 5.2. Hypereosinophilic Syndrome

Hypereosinophilic syndrome (HES) is an uncommon, multisystemic and heterogenous group of disorders, first described by Hardy and Anderson in 1968 [74]. HES can divided into many subtypes (e.g., eosinophilic granulomatosis with polyangiitis overlap, myeloid variant, lymphocytic variant, single-organ, idiopathic) [75]. Idiopathic HES is defined by the presence of: (i) long-lasting (>6 months) blood eosinophilia (>1.5 × 1053/L); (ii) organ damage and/or dysfunction that can be linked to eosinophilia and iii) lack of alternative root [76]. Cutaneous involvement is quite common, in up to 50% of the patients, but cutaneous lesions as the sole manifestations of HES are extremely rare and have been referred to as hypereosinophilic dermatitis [77]. HES-related cutaneous manifestations are usually pruritic, tender and erythematous papules or plaque with variable edema, that appear on the trunk and/or extremities [75]. The skin lesions might also present atrophic changes or post-inflammatory hyperpigmentation. Rarely livedo reticularis, skin and oral ulcers, bullous lesions; and erythroderma are often reported [77].

## 6. Drug-Related Eruptions

Drug-related eruptions are a broad group of adverse reactions to various drugs that target the skin. The possible clinical manifestations are polymorphic and non-specific for the trigger drug.

### 6.1. Iatrogenic Rash

Exanthematous eruptions are the most frequent adverse drug reactions (ADV) affecting the skin. They are often referred to as maculopapular drug eruptions or drug rashes [78]. Most drug classes can induce this type of ADV, but in the majority of cases aminopenicillins, allopurinol, sulfonamides, cephalosporins, non-steroidal anti-inflammatory drugs and aromatic anticonvulsants are involved [79].

Iatrogenic eruption presents clinically as erythematous macules and/or papules that begin on the trunk and spread symmetrically to the extremities; over time they can become confluent (Figure 4).

On the lower extremities, due to dependency, the skin manifestations can become purpuric or petechial. The rash can resemble measles and/or scarlet fever. The lesions usually appear 5–14 days after the administration of the culprit drug, but tend to appear earlier in the case of rechallenge. Iatrogenic eruption can present as an isolated finding or in association with pruritus and a low-grade fever [80]; mucous membranes are typically spared. Resolution may occur spontaneously over a period of one to two weeks after the culprit drug has been discontinued; desquamation and post-inflammatory hyperpigmentation are common sequelae [80].

Histopathologically speaking, nonspecific findings are typically seen, such as a mild superficial perivascular and interstitial lymphocytic infiltrate that may contain eosinophils in addition to interface changes.

Iatrogenic rash may present with urticarial lesions as well. As a rule, drugs are thought to be responsible for <10% of all cases of urticaria and they are more often associated with acute rather than chronic urticaria. The clinical features of drug-induced urticaria are indistinguishable from other causes and consist of itchy wheals that last less than 24 h.

Immunologic and non-immunologic mechanisms are both possible; the former is mediated by IgE antibodies, the latter, which is called anaphylactoid reaction, is due to non-immunologic histamine release. Drugs that most frequently produce immunologically-based urticaria are antibiotics such as penicillins and cephalosporins while acetylsalicylic acid is the drug that is classically involved in anaphylactoid reaction [81].

### 6.2. Fixed Drug Eruption

Fixed drug eruption (FDE) is an adverse drug reaction that can occur at all ages, but it most commonly affects adults, with reported median ages fluctuating between 35 and 60 [82]. A skin rash appears at the same anatomical location each time the culprit drug, more frequently antibiotics, analgesics, antiphlogistic drugs and hypnotics [82], is ingested/administered. Interestingly, a few cases of FDE following SARS-CoV2 vaccines (both mRNA and inactive) have recently been reported [83,84,85].

FDE usually shows as a well-demarcated round/oval patch of a few centimeters in diameter, red to purple in color with a dusky center. Post-inflammatory hyperpigmentation typically remains (Figure 5).

Less commonly, the lesions can show as blisters and/or bullae that evolve in erosions after breaking; in another less common variant, lesions are not pigmented and tend to heal without residual pigmentation [86]. Following every exposure to the offending drug, the lesions can become bigger and the number of anatomical sites involved can increase. The most reported sites of FDE include the upper and lower limbs, trunk, hands, head, lips and mucosal surfaces, with high variability depending on the study [82,86].

FDE can be also show multiple lesions, more often scattered (4 or 5 lesions) and more rarely generalized. The time lapse between drug exposure and the onset of skin manifestations can be as long as 14 days. Most patients develop the eruption within two days of exposure, especially after repeated intake of the drug [82].

From a hystopathological point of view, FDE is characterized by: (i) vacuolar interface dermatitis; (ii) superficial and deep perivascular polymorphic infiltration; (iii) individual necrotic keratinocytes that can be found in the epidermis; (iv) pigment incontinence [86].

## 7. Other Inflammatory Conditions

In this section we present a heterogeneous group of cutaneous inflammatory conditions that can be put in differential diagnosis with urticaria.

### 7.1. Wells Syndrome

Eosinophilic cellulitis, also known as Wells syndrome, is a rare inflammatory dermatosis first described by Wells in 1971 [87]. It is characterized by a benign but recurrent evolution. It most commonly occurs in adults even though pediatric cases are reported. It most commonly occurs in adults even though pediatric case are reported [88]. Etiopathogenesis is still unclear but the accumulation of eosinophils induced by T-cells producing IL-5 in response to different stimuli is hypothesized [89].

Clinically, Wells syndrome is characterized by the abrupt onset of erythematous, edematous and infiltrated plaques whose surface is sometimes covered by vesicles or bullae (Figure 6a,b). 

Skin lesions typically arise on the extremities and initially resemble urticaria or erysipelas/cellulitis but do not improve with antibiotic therapy [90]. They may be preceded by a burning sensation. After the initial eruption, the lesions evolve, presenting with a granulomatous or morphea-like appearance; the center heals while the border becomes purple [91]. The skin manifestations can be single or multiple. In the adult population, the most frequent clinical presentation is erythematous annular lesions resembling annular granuloma, while in the pediatric population the classic plaque-type variant predominates [92]. Recurrence is the rule, with a variable period of time between episodes [91].

From a histological point of view, three phases can be recognized: acute, subacute and regressive. The acute phase presents edema of the superficial dermis with a dense eosinophilic infiltrate without vasculitis. The subacute phase is characterized by the presence of flame shaped areas in the dermis, due to eosinophil degeneration of collagen fibers caused by toxic degranulation products released by eosinophils. Finally, the regressive phase shows the disappearance of eosinophils and the appearance of giant cells around collagen deposits, forming microgranulomas [92].

### 7.2. Sweet Syndrome

Acute febrile neutrophilic dermatosis, also known as Sweet syndrome (SS) is a rare inflammatory skin disorder characterized by the appearance of erythematous papules, plaques or nodules, accompanied by fever and neutrophilic leukocytosis. 

Mucosal involvement is exceptional. SS can be classified into classic or idiopathic, malignancy-associated SS and drug-induced SS. The true incidence of SS in the general population is still unknown [93], but classical SS predominates in the female populations between the ages of 30 and 60 [94].

Idiopathic SS is characterized by the sudden onset of painful, elevated, edematous, well-demarked red or purple-red papules or plaques, whose size varies from 1 to several centimeters. Sometimes the lesions may exhibit pseudovesiculation (Figure 7) or erythema multiforme-like appearance [94]. The lesions most often appear on the head, neck, trunk and upper limbs.

Bullous, pustular and subcutaneous SS are further clinical subtypes of the disease. Skin manifestations are usually associated with fever and neutrophilia (with blood polymorphonuclear leukocyte level greater than 10,000/mm^3^.)

The second variant of SS is malignancy-associated SS. Several authors reported a strong association with blood malignancies such as acute myelogenous leukemia [95]. In this form, skin lesions are less discrete; mucosal involvement is possible while blood neutrophilia is not a constant [96].

Lastly, SS can be caused by the use of medications. Various reports suggest an association between SS and the use of granulocyte colony-stimulating factor, all-trans retinoic acid and azathioprine [97].

Histopathological examination reveals a dense neutrophilic infiltrate in the upper dermis, without signs of vasculitis. The neutrophilic infiltrate may extend to the subcutaneous tissue with septal and/or lobular involvement [98].

### 7.3. Polymorphic Eruption of Pregnancy

Polymorphic eruption of pregnancy (PEP) is the most common pregnancy-specific dermatosis, with an incidence of 1 in 160 pregnancies [99]. It was first described in 1979 and it is also known in American English as pruritic urticarial papules and plaques of pregnancy (PUPPP) [100].

PEP typically occurs in the primigravida in the third trimester and only exceptionally it first appears post-partum [99]. PEP is generally regarded as a benign self-limiting inflammatory disorder, unassociated with fetal or maternal morbidity and mortality. Clinically speaking, PEP is characterized by the rapid onset of extremely pruritic erythematous and edematous papules of 1–2 mm that can coalesce to form plaques [101]. The eruption is usually on the abdomen, often within the striae distensae (Figure 8); sparing of the umbilical area is typical. 

It can spread to the upper thighs, buttocks and the back with facial sparing and infrequent hand or foot lesions [102]. In addition to this symmetric presentation with plaques and papules, vesicles, target lesions, and annular wheals have also been reported. Some authors classify PEP according to the clinical presentation [101]. One of these classifications divides PEP in three categories: type I with urticarial papules and plaques, type II with non-urticarial erythema, papules or vesicles and type III with a combination of the two forms aforementioned.

### 7.4. Erythema Multiforme

Erythema multiforme (EM) is an immune-mediate condition in response to various triggers. Infections are associated with 90% of cases of EM and the most common pathogens in adults are herpes simplex virus (HSV) type 1 and *Mycoplasma pneumoniae* [103]. In children infections are once again the most common cause; together with HSV-1 and *Mycoplasma Pneumoniae*, other pathogens deserve a mention: COVID19, Epstein-Barr virus, cytomegalovirus, vulvovaginal candidiasis [104,105].

Drugs cause less than 10% of EM cases, however many classes of drugs have been associated with EM, most commonly non-steroidal anti-inflammatory drugs, antiepileptics, and antibiotics [106].

Clinically, EM is characterized by pink or red papules, that can coalesce into edematous plaques. In a few days the plaques transform into the classic target or iris lesion, which typically develops three concentric segments: a dark center and two surrounding rings, a lighter pink one and a peripherical red ring. Vesicles may be present, especially in the central portion (Figure 9). These lesions can cause burning or itching. 

The cutaneous manifestations are found symmetrically on the extremities, especially on extensor surfaces and then spread centripetally. Palms and soles may be involved. The lesions usually heal without scarring, but hyperpigmentation may occur [107].

Mucosal involvement is present in 25% to 60% of patients with EM, and usually begins with edematous, erythematous lesions that may develop into shallow erosions with pseudo-membranes. 

A biopsy is not necessary for diagnosis if the clinical features are clear. The histological findings vary, depending on the morphology and duration of the lesion on which the biopsy is performed. In the early stages, perivascular mononuclear cell infiltrate can be seen. Biopsy of a target lesion may show pronounced dermal edema peripherically, while necrotic keratinocytes or epidermal changes occur centrally [108]. Direct immunofluorescence (DIF) can differentiate between EM and autoimmune blistering diseases, such as bullous pemphigoid.

Recently a few authors described cases of EM or EM-like eruptions linked to COVID-19 [109].

### 7.5. Hyper IgE Syndromes

Hyper IgE syndromes (HIES) are primary immunodeficiency disorders characterized by: (i) eczema/atopic dermatitis; (ii) recurrent cutaenous and pulmonary infections; (iii) elevated IgE [110]. Skeletal and connective tissue complaints are further characteristic features.

Two types of HIES exist. Type I is a hyper-IgE syndrome with autosomal dominant transmission, characterized by abnormalities of the immune, skeletal and vascular systems with connective tissue alterations and more; type II is autosomal recessive and usually not have musculoskeletal abnormalities [111].

Patients with hyper IgE syndrome show cutaneous manifestations similar to atopic dermatitis, with neonatal onset in more than 50% of cases [112]. Despite extremely high serum IgE levels and eosinophilia, urticaria as well as other allergic manifestations, such as allergic rhinitis, asthma, and anaphylaxis, are quite unusual. 

The major causal determinant of the hyper-IgE syndrome are dominant negative variants in the STAT3 gene [112].

### 7.6. Insect Bite Reactions

Insects are living creatures within the arthropods’ group. They can cause lesions through biting or stinging. A bite is a wound produced by the mouth the animal while a sting is inflicted through a special structure that can inject venom.

Insect bites and stings are spread worldwide; in temperate climates they are a seasonal phenomenon, but indoor infestation can persist all-year-long.

Arthropods produce a wide spectrum of clinical lesions. Usually, insect bites cause the development of erythematous urticarial papules of 2–10 mm in diameter. These lesions can be grouped or disseminated, are pruritic, and are often excoriated (Figure 10).

Occasionally, vesiculobullous reaction, targetoid lesions, edematous plaques or purpuric papules may develop. Staphylococcal secondary infection of the lesions frequently occurs [113]. Insect bites involve the exposed areas of the head, neck, and extremities but the development of generalized manifestations such as papular urticaria is a possibility [114]. Stings from bees, wasps and ants produce an immediate sensation of burning and pain, followed by the forming of local erythema and swelling [113]. Insect bites resolve in 5–10 days; resolving lesions are often hyper-pigmented.

## 8. Conclusions

The aim of this review is to provide clinicians a useful tool for correctly suspecting and identifying the principal skin disorders that can be misdiagnosed as urticaria. These conditions must be taken in consideration in case of atypical wheals, in morphology and duration (>24 h), association with other elementary skin lesions, absence of angioedema and appearance of systemic symptoms. The presence of one or more of these features should prompt further investigations in order to allow an accurate diagnosis and consequently ensure the best therapeutic choice for the patient.

## Figures and Tables

**Figure 1 biomedicines-11-01096-f001:**
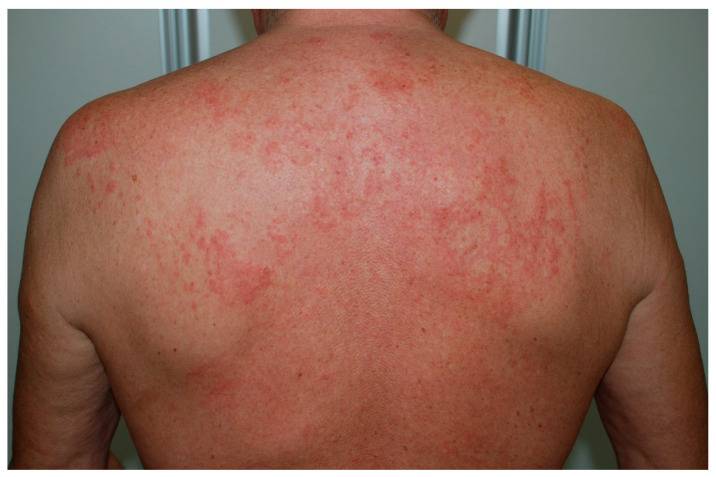
Wheal-and-flare papules and plaques on the back due to urticarial vasculitis.

**Figure 2 biomedicines-11-01096-f002:**
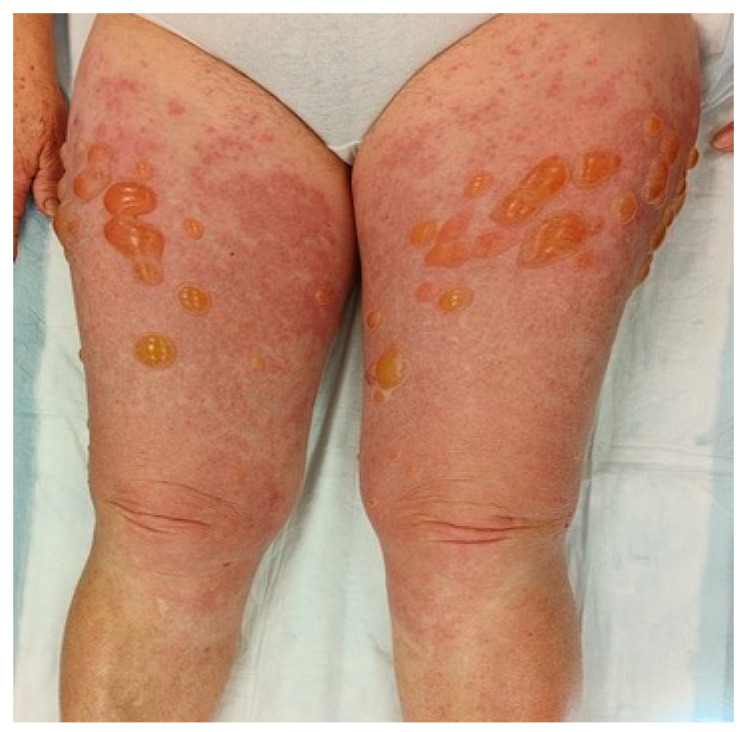
Clinical presentation of bullous pemphigoid showing tense bullae with a clear content overlying wheal-and-flare plaques on lower limbs.

**Figure 3 biomedicines-11-01096-f003:**
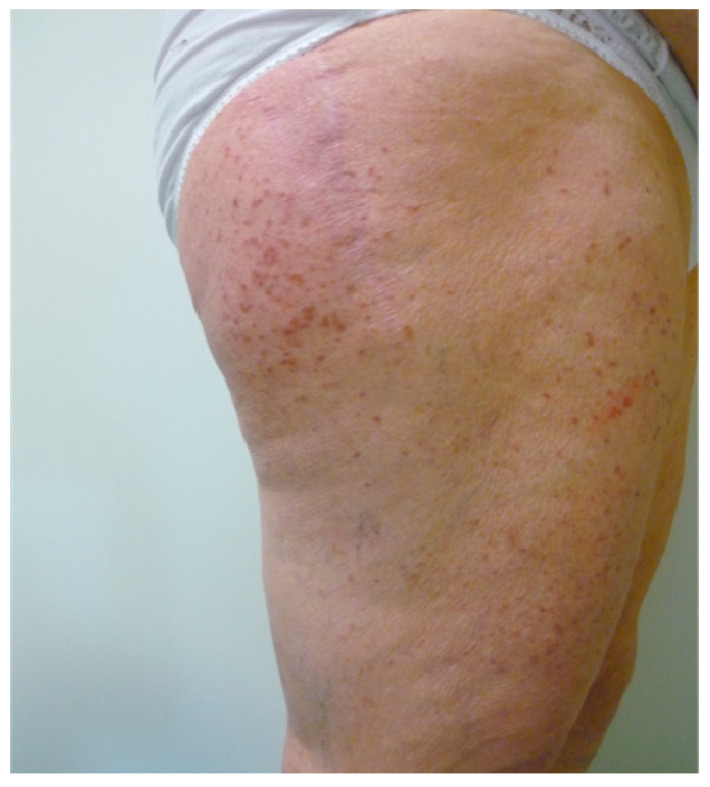
Yellowish to deep brown macules and papules on thighs in an old woman affected by mastocytosis.

**Figure 4 biomedicines-11-01096-f004:**
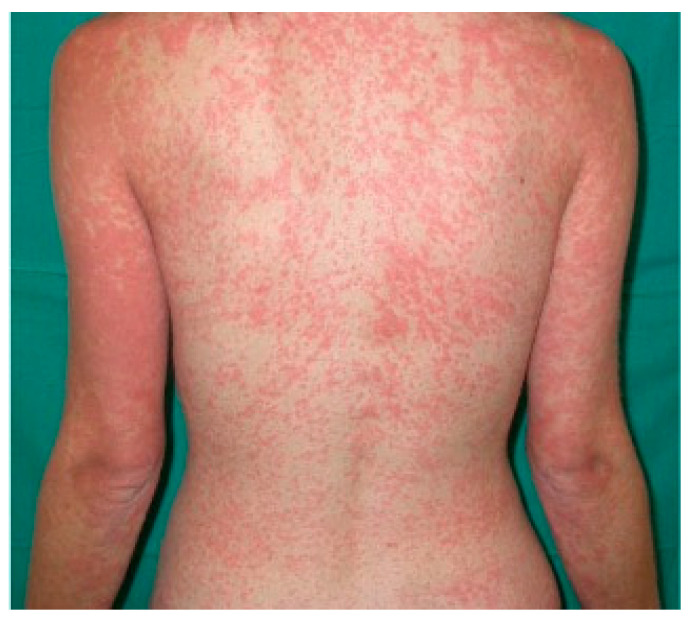
Drug-related erythematous maculopapular eruption on trunk and upper extremities in a young girl.

**Figure 5 biomedicines-11-01096-f005:**
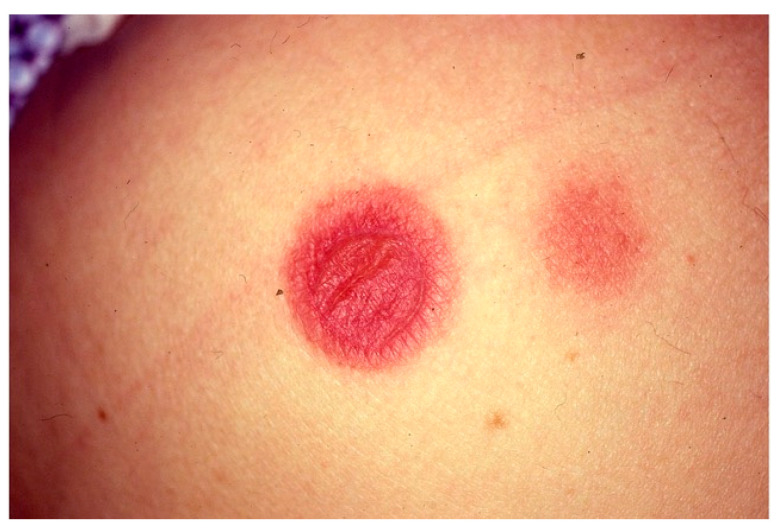
Clinical manifestations of fixed drug eruption characterized by sharply demarcated erythematous-violaceous and oval patches.

**Figure 6 biomedicines-11-01096-f006:**
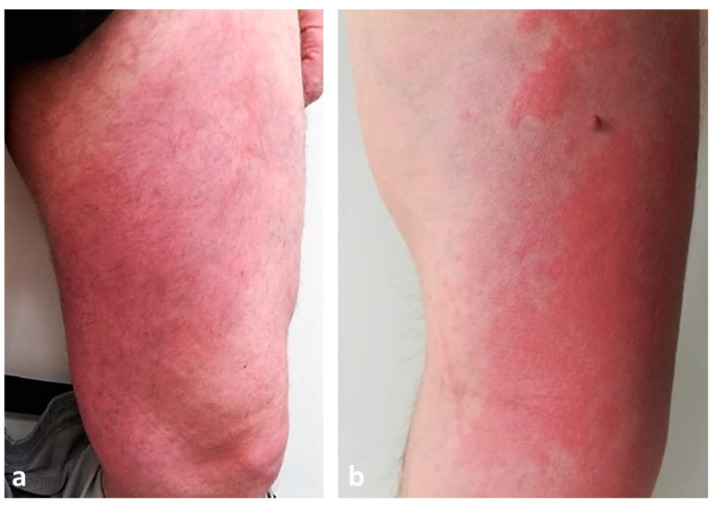
Clinical presentation of Wells syndrome showing erythematous, edematous and infiltrated plaques on lower (**a**) and upper limbs (**b**).

**Figure 7 biomedicines-11-01096-f007:**
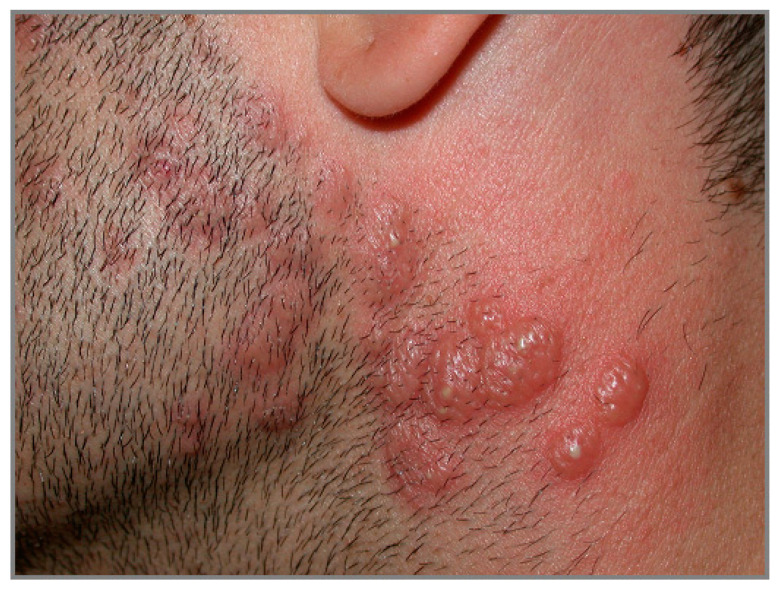
Tender erythematous papules and nodules associated with pseudovesiculation and pustules on head and neck in a man affected by Sweet syndrome.

**Figure 8 biomedicines-11-01096-f008:**
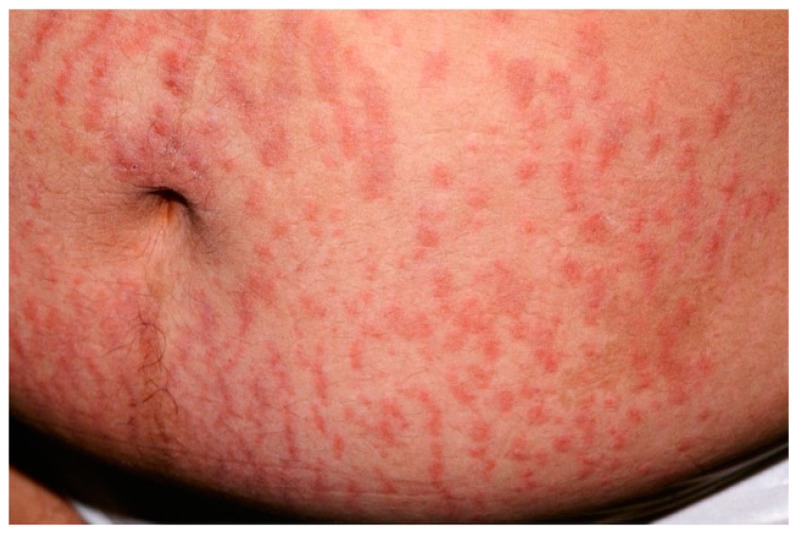
Clinical presentation of polymorphic eruption of pregnancy showing erythematous and edematous papules on a pregnant abdomen and within the striae distensae.

**Figure 9 biomedicines-11-01096-f009:**
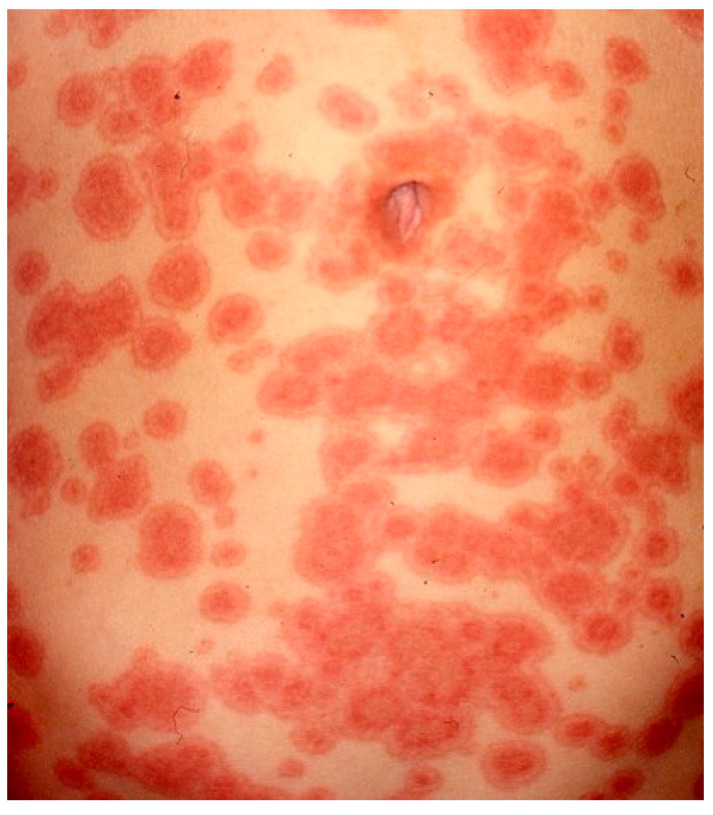
Abdominal multiple target lesions due to erythema multiforme.

**Figure 10 biomedicines-11-01096-f010:**
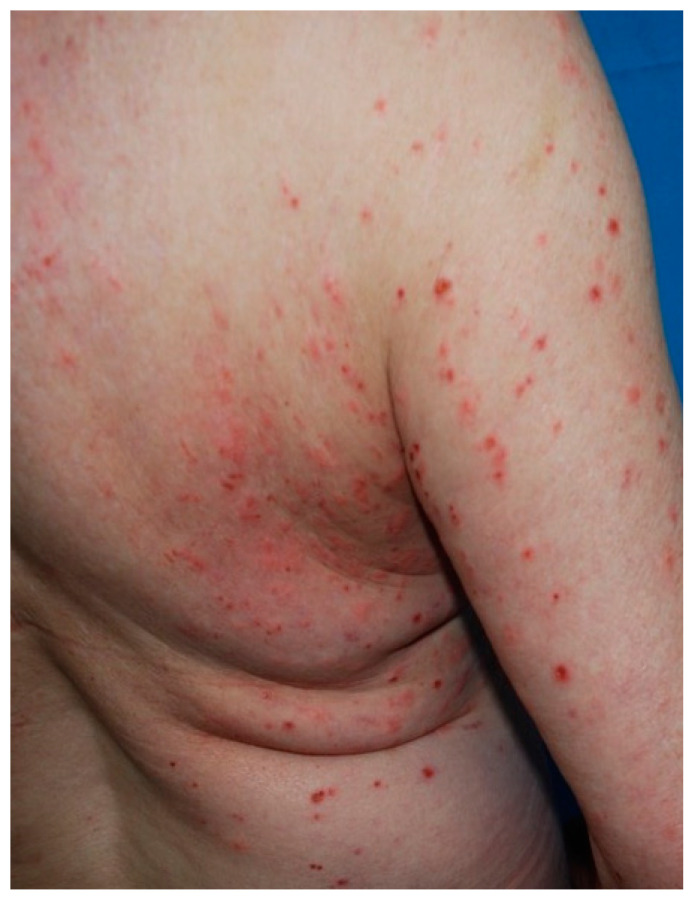
Multiple urticarial and excoriated papules due to insect bite reactions.

**Table 1 biomedicines-11-01096-t001:** Main differential diagnoses of urticaria.

Disease	Epidemiology	Clinical Features	Histopathology
Urticaria	All ages	Wheals < 24 h, angioedema	Swelling of superficial dermis, dilation of vessels, perivascular inflammatory infiltrate
Urticarial Dermatitis	Mean age: 60 y.o.	Overlapping urticarial and eczematous features—simultaneously or sequentially	Dermal hypersensitivity reaction pattern: dermal and perivascular lymphocytic infiltrate + scattered eosinophils. Epidermal spongiosis. Corneum unaffected.
Urticarial vasculitis	Mean age: 30–40 y.o.	Wheals (0.5–5 cm) lasting >24 h, dark-red or brown. Painful and/or burning sensation.	Karyorrhexis + erythrocytes outside vessels + fibrin inside vessels + neutrophilic infiltrate
Cutaneous lupus erythematosus	Females 3rd–4th decade	Broad spectrum. LE specific and LE nonspecific skin manifestations.	overlapping features: (i) epidermal vacuolization + apoptosis, (ii) interface dermatitis, (iii) dermal inflammation, (iv) follicular plugging, (v) thick basal membrane (vi) dermal mucin deposits
Bullous Pemphigoid	Elderly	Polymorphic Tense bullae → erosions/crusts on erythematous urticarial skin	(i) subepidermal rupture (ii) perivascular infiltrate of eosinophils, (iii) eosinophilic spongiosis. DIF: linear deposition of C3 and IgG at the basal membrane
Pemphigoid gestationis	Pregnant women/post-partum	Polymorphic: eczematous, erythema multiforme-like, urticaria-like, papules → tense blisters. Starts on the abdomen	(i) papillary edema, (ii) dermal inflammatory infiltrate, (iii) subepidermal blistering. DIF: linear deposition of C and IgG at the dermal-epidermal junction.
Dermatitis herpetiformis	Mean age: 30–40 y.o.	Papulo-vesicular rash at the extensor surfaces of the limbs	Vesicle = sub-epidermal split + neutrophilic infiltrate at dermal papillae. Perivascular polymorphic inflammatory infiltrate
Autoimmune progesterone dermatitis	Women of childbearing age	Heterogenous: urticaria-like, vesiculobullous, erythema multiforme, eczema, maculopapular eruption, purpura/petechiae	Nonspecific findings
Mastocytosis	Different onset age depending on subtype	MPCM: yellowish/brown lesions. TMEP: multiple erythematous/hyperpigmented macules + telangiectasias. Cutaneous mastocytoma: solitary yellowish/brown macule/nodule/plaque. DCM: generalized erythema and pachydermia.	Multifocal infiltrates of mast cells
Hypereosinophilic syndrome	All ages	Pruritic, tender, erythematous papules/plaques. Trunk and/or extremities.	Nonspecific findings
Iatrogenic rash	All ages	Erythematous macules/papules → confluent. Start on the trunk → extremities symmetrically.	Nonspecific findings
Fixed drug eruption	All ages	Red/purple, well-demarcated round/oval patch	(i) vacuolar interface dermatitis (ii) polymorphic infiltration (iii) necrotic keratinocytes (epidermis) (iv) pigment incontinence
Wells syndrome	Adults	Erythematous, edematous, infiltrated plaques. Abrupt onset.	Acute: dermal edema + dense eosinophilic infiltrate without vasculitis. Subacute: flame figures (degeneration of collagen due to eosinophil degranulation) Regressive: no eosinophils + microgranulomas
Sweet syndrome	All ages	Painful, elevated, edematous, well-demarked red/purple-red papules/plaques	dense neutrophilic infiltrate in the dermis, without vasculitis
Polymorphic eruption of pregnancy	Pregnant women	Erythematous and edematous papules → plaques. Starts on the abdomen.	Nonspecific findings
Hyper IgE syndromes	50% neonatal onset	Eczema/atopic dermatitis	Nonspecific findings
Insect bite reactions	Any age	Erythematous urticarial papules (2–10 mm)	Nonspecific findings
Neutrophilic urticarial dermatosis		Urticarial rash	Intense neutrophilic infiltrate
Erythema multiforme	All ages	Pink/red papules, → plaques → classic target lesion	Early stages: perivascular mononuclear cell infiltrate. Target lesion: dermal edema (peripherically) + necrotic keratinocytes/epidermal changes (centrally)

DIF: direct immunofluorescence, LE: lupus erythematosus, MPCM: maculopapular cutaneous mastocytosis, TMEP: teleangectasia macularis eruptiva perstans, DCM: diffuse cutaneous mastocytosis.

**Table 2 biomedicines-11-01096-t002:** Principle autoinflammatory urticarial syndromes.

Syndrome	Age of Onset	Duration of Clinical Manifestation	Skin Lesions	Systemic Symptoms
Familial cold auto-inflammatory syndrome	Infancy	<24 h	Cold-induced urticarial eruptions	Fever, arthralgia, nausea, vomiting and conjunctivitis
Muckle-Wells syndrome	Infancy-adolescence	1 to 3 days	Wheal-and-flare macules and papules	Fever, arthralgia, conjunctivitis, progressive sensorineural loss and risk of amyloidosis
Chronic infantile neurologic, cutaneous and articular syndrome	Neonatal period	Continuous with flares	Persistent urticaria-like rash	Fever, neurologic disorders (aseptic meningitis), ocular manifestations (progressive visual loss due to papilledema and atrophy of optic nerve) and joint involvement (deforming osteo-arthropaty)
Schnitzler’s syndrome	Adulthood (mean age 50 years)	12 to 36 h	Slightly itchy wheals occasionally with hyperpigmented evolution	Recurrent fever, fatigue, general malaise, arthralgia, myalgia, bone pain, hepato-splenomegaly and lymphadenopaty. IgM or IgG monoclonal gammopathy as obligate criterion for diagnosis.
Adult-onset Still’s disease	16–35 years	24–36 h	Mildly itchy skin rash with salmon-pink macules and papules	High-spiking fever in the afternoon or early evening, arthralgia or arthritis, sore throat, serositis, hepato-splenomegaly and lymphadenopaty
Systemic-onset juvenile idiopathic arthritis	<16 years	24–36 h	Transient pink macular eruption	High-spiking fever, poly-arthritis, serositis, hepato-splenomegaly and lymphadenopaty
Mevalonate kinase deficiency	First year of life (mean age 6 months) in HIDS and at birth or ante-natal in mevalonic aciduria	3–7 days	Evanescent maculo-papular rashes occasionally with purpuric aspects	Recurrent fever, abdominal pain, vomiting and/or diarrhea, lymphadenopathy, arthralgia and non-erosive arthritis in HIDS. Neurologic alterations and hematologic abnormalities in mevalonic aciduria. Elevated IgD levels as laboratoristic finding.
TNF-receptor-associated periodic syndrome	<10 years	5 to 25 days with recurrences every 4 to 6 weeks	Migrating erythema with centrifugal evolution and less frequently urticarial plaques	High fever, abdominal and chest pain, arthralgia, myalgia, eye involvement (uveitis or conjunctivitis)

## Data Availability

Not applicable.

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
