# Peer review of "Urticaria: A Narrative Overview of Differential Diagnosis"

_biomedicines, 2023, doi:10.3390/biomedicines11041096_

Round 1
Reviewer 1 Report
INTRODUCTION
- At the end of the first paragraph, the authors should provide some epidemiological background for chronic urticaria, which can be observed both in adults (refer to: Prevalence estimates for chronic urticaria in the United States: A sex- and age-adjusted population analysis. J Am Acad Dermatol. 2019 Jul;81(1):152-156. doi: 10.1016/j.jaad.2019.02.064) and children .
- At the end of the introduction, the authors should clearly explain the aim of this review.
MATERIALS AND METHODS
- This section is not appropriate, since this is a narrative review. So, I recommend to remove it.
GENERAL
- Following the previous comments, the authors should re-organize this review in sections and subsections, which should be appropriately numbered.
- The different types of dermatological disorders should be grouped in each section, according to the conventional classification. In the current version, there is no clear order.
- The authors included several figures in this paper. Are these original figures of authors’ property? Please, clarify this. If so, did the authors obtain the informed consent for publication use from their patients? Please, clarify this point.
- Each subsection (describing a specific disorder) should follow a logical order to describe the condition (e.g. etiology, pathogenesis, histopathology, clinical aspects, prognosis, therapy). This order should be applied to each subsection/specific disease.
- The length of each subsection should be overall uniform (of course, with some major details for those disorders having a higher epidemiological burden)
- An additional chapter focused on the classification of chronic urticaria with detailed explanation should be added (Urticaria and Angioedema: an Update on Classification and Pathogenesis. Clin Rev Allergy Immunol. 2018 Feb;54(1):88-101. doi: 10.1007/s12016-017-8628-1). Moreover, information about the general therapeutic approach should be also included, considering that the core topic of the review is (chronic) urticaria and its differential diagnosis. In detail a few concept about omalizumab (Monoclonal Antibodies in Treating Chronic Spontaneous Urticaria: New Drugs for an Old Disease. J Clin Med. 2022 Jul 30;11(15):4453. doi: 10.3390/jcm11154453) and its mechanisms are needed (see: Potential biomarkers in asthma and chronic spontaneous urticaria. Cell Immunol. 2020 Dec;358:104215. doi: 10.1016/j.cellimm.2020.104215)
ERYTHEMA MULTIFORME
- “Infections are associated with 90% of cases of EM and the most common pathogens are herpes simplex virus (HSV) type 1 and Mycoplasma pneumoniae, especially in children.” The authors correctly mentioned this aspect. However, this statement should be supported by appropriate references for Mycoplasma pneumoniae (see: Is There any Relationship Between Extra-Pulmonary Manifestations of Mycoplasma Pneumoniae Infection and Atopy/Respiratory Allergy in Children? Pediatr Rep. 2016 Mar 31;8(1):6395. doi: 10.4081/pr.2016.6395).
- Conversely, I have some concerns about highlighting the role of HSV-1 in pediatric erythema multiforme, whereas other Herpesviridae are more frequently involved (e.g. CMV, EBV, or others). Please, revise this point and support it with appropriate reference.
CONCLUSION
- The authors should summarize the main learning and practical points. This is not the section to repeat the aim of this revies. In detail, the novel contribution to the scientific literature should be highlighted and clearly explained.
REFERENCES
- The references should be completed and updated according to the re-organization of the manuscript and the previous recommendations.
FIGURES
- please, see comments below. If appropriate for publications, some figures could be improved (Figure 5 could exclude the head; Figure 4 could be conversely expanded to clarify the body region, etc.)
TABLES
- There are no tables.
- I would suggest adding one schematically summarizing all the differential diagnosis and the main aspect.
- Moreover, I suggest adding a table focused on autoinflammatory syndromes, where urticarial rashes are important diagnostic criteria. Here, the authors could provide additional details on the periodic symptoms and main characteristics.
Reviewer 2 Report
This is an interesting style of writing that is engaging.
1. It is difficult for me to see how including Sweet's syndrome into this review has meaning. I could not find a single PubMed-referenced reference for urticarial Sweet's syndrome.
2. Iatrogenic rash is discussed as a topic that is not related at all to urticarial drug eruptions.
3. Conditions like fixed drug, erythema multiforme, lupus, bullous pemphigoid, early in the course may be urticarial, but they evolve to other morphologies. The authors do not make this clear.
Other urticarial disease can be brought to mind such as polymorphic eruption of pregnancy, pemphigoid gestationis and others. Why were these not mentioned?
Reviewer 3 Report
This is a well-written overview of differential diagnoses to urticaria in a review form.
I have only a minor comment: When presenting a review of the literature a flow chart and criteria for choosing or excluding papers could be useful. You mention that you included papers published until December 2022. One might expect that much more than 90 papers were found. If you only chose some, please give more detail about the selection criteria. If you only included the most recent you could mention this, instead of claiming the whole body of literature until present has been read through. May be misleading.
Round 2
Reviewer 1 Report
Overall, the authors addressed most of my comments appropriately. However, as stated by the authors this is a narrative review and, therefore, the material and methods section makes no sense. I still recommend the authors to remove it. Anyway, the information provided is very general and, despite the authors' comment, I think it does not add anything to this manuscript, in my opinion.
Author Response
We thank the reviewer for their observation.
We removed the material and methods section as suggested.
Reviewer 2 Report
The authors have addressed the major criticisms.
Author Response
We thank the reviewer for their observation and for appreciating the revised version of our manuscript.